# Sexual Differences in *Eurya loquaiana* Dunn Floral Scent and How Pollinators Respond

**DOI:** 10.3390/plants11192560

**Published:** 2022-09-28

**Authors:** Qian Wang, Bo Ding, Hongping Deng

**Affiliations:** 1Chongqing Key Laboratory of Plant Resource Conservation and Germplasm Innovation, Institute of Resources Botany, School of Life Sciences, Southwest University, Chongqing 400715, China; 2Biotechnology Research Center, Southwest University, Chongqing 400715, China; 3Chongqing Research Center for Low Carbon and Ecological Environment, Chongqing Academy of Science & Technology, Chongqing 401123, China

**Keywords:** *Eurya*, volatile organic compound, SPME-GC-MS, floral scent, subdioecious

## Abstract

*Eurya* plants are usually dioecious or subdioecious with small fragrant flowers. Here, we investigate the floral scent components of the subdioecious species *Eurya loquaiana* Dunn and how floral scent affects pollinators. Headspace solid-phase microextraction–gas chromatography–mass spectrometry (HS-SPME-GC-MS) was used to compare the floral scents of male, female, and hermaphrodite flowers. We also test whether differences in floral scent affect the foraging behaviors of pollinators and describe the flower morphological traits of the three sexes. Twenty-eight floral scent compounds were tentatively identified, and four monoterpenoids were tentatively identified as the most abundant compounds: linalool oxide (pyranoid), linalool, lilac aldehyde, and linalool oxide (furanoid). There were floral scent differences among the sex types, and male flowers were more attractive to pollinators in the wild, even when visual factors were excluded, indicating that pollinators likely distinguish sexual differences by floral scent. In the competition for pollinators, the advantage that male flowers have over female and hermaphrodite flowers can likely be accounted for the differences in floral scent and display size.

## 1. Introduction

The genus *Eurya* Thunberg (Pentaphylacaceae) contains ~130 species [1,2,3]. The plants of this genus are usually evergreen shrubs with small fragrant flowers, and they have been named mountain osmanthus [4]. These species are common among the undergrowth in evergreen broadleaved forests, have some ornamental and medicinal values [2,5,6,7], and are an important nectar and pollen source for the survival of bees in the winter in south China [4]. Though the species of the genus *Eurya* are not common aromatic plants, they show promise as a source of essential oil.

Most species in this genus are described as dioecy, in which male and female functions are separated between different individuals [2,3]. However, subdioecy has also been reported in a few *Eurya* species, with rare bisexual individuals coexisting with male and female plants in the same population [8,9,10,11,12]. Sex differentiation is often accompanied by the differentiation of multiple traits (i.e., perianth size, nectar volume and content, and floral scent) [13,14,15]. Pollinators might perceive these differences, causing them to discriminate during visitations. Studies addressing the sexual differences of flowers and how these affect pollinators could provide an important reference for the study of sexual evolution, especially in subdioecious species, which are often regarded as a transitional state in sexual evolution [10].

Floral scents are composed of a huge variety of volatile compounds that play important roles in attracting pollinators or in deterring herbivores [16]. The compositions and concentrations of floral scents affect the interaction between plants and visitors [16,17]. Moreover, scent patterns might correspond to visual patterns within a flower to enhance attractiveness to pollinators [18]. Sexual dimorphism in scent has been studied in various sexually dimorphic species, and the differences between sexes are usually characterized by scent composition and emission rate [15]. Ashman et al., (2005) found that the pollinator preference of hermaphrodite in *Fragaria virginiana* was related to the scent released by the flowers, and this discrimination was primarily due to the scent of hermaphrodite anthers [19]. Waelti et al., (2009) found that male flowers emitted significantly larger amounts of scent than female flowers, and the male moths (pollinator) showed significant preference of male over female flowers in *Silene latifolia* [20]. Okamoto et al., (2013) found that *Epicephala*-pollinated Phyllanthaceae plants consistently exhibit major qualitative differences in scent between male and female flowers. In their choice test, the mated female *E. bipollenella* preferred the scent of male flowers over *Glochidion zeylanicum* female flowers, suggesting the male floral scent elicits pollen-collecting behavior. However, less attention has been paid to the sexual differentiation of floral scent in subdioecious species [21].

Despite their fragrant flowers, few studies have addressed the floral scent compositions of the *Eurya* species [22,23]. To better understand the floral scent of the genus *Eurya* and its ecological role in pollination, in this study, we investigate the floral scent components of *E. loquaiana,* using headspace–solid-phase micro-extraction (HS-SPME) combined with gas-chromatography–mass-spectrometry (GC-MS) [24]. We also describe flower morphological traits and investigate how floral scent might affect the foraging behavior of pollinators.

## 2. Results

### 2.1. Floral Morphology and Rewards

Obvious differences in flower morphology were detected among the male, female, and hermaphrodite flowers of *E. loquaiana* (Figure 1, Table 1). The petals of male flowers were 1.1–1.4-times larger and ~1.5-times heavier than female and hermaphrodite flowers (Table 1), and these differences were statistically significant (Table 1). The size and weight of female and hermaphrodite flowers were similar, but the petals of hermaphrodite flowers were wider than those of female flowers (Table 1). Female and male flowers offer different rewards to the pollinators, with the male flowers producing only pollen and the female flowers producing only nectar. Hermaphrodite flowers can produce both nectar and pollen, but the yields are less than those of the unisexual flowers (Table 1).

### 2.2. Tentative Identification of Scent Components

Twenty-eight compounds were tentatively identified in *E. loquaiana* flowers with differences sexes, including thirteen aliphatic hydrocarbons, eleven terpenoids, and four benzenoids (Table 2). The relative content of terpenoids accounted for more than 75% of the identified compounds. The most abundant compounds were identified as four monoterpenoids: linalool oxide (pyranoid), linalool, lilac aldehyde, and linalool oxide (furanoid), of which the relative content was higher than 8% in all kinds of flowers. Meanwhile, the relative content of linalool oxide (pyranoid) and linalool were higher than 20% in all kinds of flowers.

### 2.3. Comparison of the Floral Scent among the Sexes of Eurya loquaiana

Although the same compounds were detected in all sex types, their compositions varied (Table 2). The female floral scent contained a greater proportion of terpenoids (86.9%) than the male (81.9%) and hermaphrodite (79.0%) floral scents. Aliphatic hydrocarbons accounted for 19.6% of the compounds identified in the hermaphrodite floral scent, which is greater than in the male (16.0%) and the female (11.6%) floral scents. Regarding the four most abundant compounds, females contained a significantly higher percentage of linalool oxide (pyranoid) than hermaphrodites; females and males contained a significantly higher percentage of lilac aldehyde and linalool oxide (furanoid) than hermaphrodites; the three sex types contained equivalent amounts of linalool (Table 2). The contents of the four main substances were estimated using 2-methyl-3-heptanone as an internal standard, and the highest content all appeared in male flowers (Figure 2).

The results of the NMDS analysis of floral scents of the different sexes are shown in Figure 3. The separation between the different sexes was confirmed using one-way ANOSIM (R = 0.5432, *p* = 0.001). The detailed ANOSIM comparisons showed clear separation between male, female, and hermaphrodite flowers (ANOSIM R (females vs. hermaphrodites) = 0.6444 (*p* = 0.001); ANOSIM R (males vs. hermaphrodites) = 0.7537 (*p* = 0.004)) but a relatively weak separation between male and female flowers (ANOSIM R (males vs. females) = 0.2111 (*p* = 0.053)).

### 2.4. Pollinator Behavior

In the field, the most frequent visitors of *Eurya loquaiana* were hoverflies, followed by honeybees and bumblebees (Figure 4a). In addition, insects from Vespa, Calliphoridae, and Formicidae were observed occasionally (visiting frequency was less than 0.01 insect min^−1^ plant^−^^1^). Due to the differences of body size and habit, there are some differences in flower-visiting behavior among the visitors. Bees and bumblebees often show the action of feeding nectar when visiting nectar-rich flowers (female and hermaphrodite flowers), and the action of collecting pollen when visiting pollen-rich flowers (male and hermaphrodite flowers). However, hoverflies always showed the action of searching and feeding of the nectar on all types of flowers and stayed on a flower for a long time. We observed large amounts of pollen on the bodies of bumblebees and bees, but only small amounts on the relatively smooth bodies of hoverflies. Therefore, we guess that bees and bumblebees may have higher pollination efficiency than hoverflies.

In order to explore whether a sexual preference exists at the plant level, we compared the visiting frequency among plants with different sexual types of flowers. The results showed that the visiting frequency on male plants was significantly higher than that on female and hermaphrodite plants, whether it was hoverflies, bees, or bumblebees (Figure 4b). The visiting frequency of hoverflies on male flowers was about 1.4~1.8 times that of female and hermaphrodite flowers. In honeybees and bumblebees, the multiples reached up to 3.3~5.2 and 1.9~4.3 times. In addition, hermaphrodite plants seemed to be more attractive to hoverflies than female plants. The visiting frequency on hermaphrodite flowers was about 1.4 times that of female flowers in hoverflies, whereas this difference was not observed in bees and bumblebees (Figure 4b).

In order to exclude the difference of visiting behavior caused by the floral display, we controlled the number of flowers on each branch and observed pollinator responses. The results indicated that there was no significant difference among the visiting frequency of different sexual types (Kruskal–Wallis test, *χ*^2^ = 0.738, *df* = 2, *p* = 0.69, Figure 4c), but the pollinators (mainly honeybees) tended to choose male flowers first in each round of visits. About 39.05% of the insects chose the male flowers preferentially during each round of visit, while 30.08% of female flowers and 30.87% of hermaphrodite flowers were preferred. The selective probability for male flowers was significantly higher than that for the female and hermaphrodite flowers (Kruskal–Wallis test, *χ*^2^ = 8.81, *df* = 2, *p* = 0.01, Figure 4d).

To rule out visual effects, an olfactometer test was also performed. In this test, honeybees showed a preference for the floral scent of males over females and hermaphrodites (Kruskal–Wallis test, *χ*^2^ = 28.269, *df* = 3, *p* < 0.001, Figure 4e). About 80% of the tested insects chose the direction of the male flowers.

## 3. Discussion

So far, most of the studies addressing *Eurya* plants have concentrated on morphology and anatomy; only a few studies have focused on floral scent components. Wang et al., reported two components of *E. japonica*: α-pinene and linalool [22]. Motooka et al., detected 87 compounds in essential oils from *E. japonica* flowers, and the main compounds were linalool, (9Z)-tricosene, and nonanal [50]. In this study, linalool oxide (pyranoid), linalool, lilac aldehyde, and linalool oxide (furanoid) were recognized as the main floral scent components of *E.*
*loquaiana*. These compounds exhibited a common floral odor that resembles lavender, citrus, or lemon [51]. Linalool and its oxides are common scent compounds in various plants that attract different pollinators, mainly various insects [16]. In winter—the flowering season of *E.*
*loquaiana*—it is often dark in the evergreen broad-leaved forests and the downward-facing axillary flowers are often shaded by leaves. Therefore, flower scent is likely to play an important role in attracting the pollinators of this species.

Sexual differentiation of floral scent might be expected in genus *Eurya*, as this difference has been observed both in *E. japonica* [22,23] and in this study. Miyazawa et al., reported the different compositions of essential oils of male and female flower buds in *E. japonica* [23], and Wang et al., compared the flower scent of *E. japonica* using a dynamic headspace method [22]. The same species was examined in these two studies; however, different sampling types lead to different results. Miyazawa et al., (2016) found that α-terpineol and 3-carene were the most abundant compounds in male oil, while (2E)-hexenal and camphor were the most abundant compounds in female oil. Wang et al., (2018) observed no qualitative differences, but males emitted significantly higher amounts of α-pinene and marginally higher amounts of linalool compared with females or hermaphrodites [22]. In our study, the same compounds were detected in male, female, and hermaphrodite flowers, but the compositions varied (Table 2, Figure 3). Male flowers seem to release more scent compounds than female and hermaphrodite flowers (Figure 2).

Sexual preference has been observed in *E. loquaiana*. The visiting frequency on male plants was significantly higher than on female and hermaphrodite plants in the field. However, this difference became not significant when the number of flowers on each branch was controlled. This suggests that the floral display could influence insect selection in *E. loquaiana*. In addition, the pollinators tended to choose male flowers first in each visit in the controlled experiment. This indicated that floral scent, visual differences (e.g., flower size, stamen color), and reward difference (e.g., nectar and pollen amount) may also influence the choice of pollinators, excepting for the floral display. The results of an olfactometer test further demonstrated the pollinator preference for the scent of male flowers. This may be caused by the differences in the composition of floral scent or the by the fact that male flowers may release more scent compounds. In this experiment, honeybees and bumblebees were the main pollinators; they have great learning abilities and might distinguish floral scent patterns and might transfer these to other corresponding patterns [18,52].

Insect preference for male flowers (or male functional flowers) has been reported in many sexually dimorphic or sexually polymorphic species [13,19,20,21]. This is the result of long-term natural selection, as female fertility requires only a few visits to fully saturate, while male fertility increases with the number of pollinator visits. There are three sex types of flowers in *E. loquaiana*. Pollinator preference for male flowers may lead to the evolution of hermaphrodite to female. This research indicates that the subdioecious system of genus *Eurya* is an intermediate stage in the evolution from hermaphroditism to dioecy and provides more details for understanding the evolutionary process [10].

## 4. Materials and Methods

### 4.1. Study Site and Study Species

This study was conducted at Jinyun Mountain natural reserve, Chongqing, China (N 29.82~29.85 E 106.3~106.5, 564~907 m). According to local meteorological data, the annual mean temperature was 13.6 °C and the annual mean precipitation was 1611.8 mm [53].

*E. loquaiana* Dunn is an evergreen broadleaf shrub that is widely distributed in south China, occurring between 400–2000 m above sea level [3]. Its flowers are insect-pollinated and bloom from October to December. Though this species was described as dioecy, the population is subdioecy at this sampling site. In seven 20 × 20 m plots at our study site, the density of *E. loquaiana* ranged from 0.06~0.17 plant/m^2^, the female–male ratio ranged from 0.77–2.33, and the proportion of hermaphrodites fluctuated widely (from 2.63% to 27.03%).

### 4.2. Floral Morphology and Rewards

To determine the morphological characteristics of the flowers, 45 flowers from 9 female plants, 55 flowers from 11 male plants, and 44 flowers from 9 hermaphrodite plants were randomly collected, then dissected and measured under a Nikon SM21000 stereo microscope. To determine the nectar volume of the flowers, branches with 10~20 mature flower buds from three female plants and three hermaphrodite plants were randomly selected, labeled, and bagged with nylon bags (80 meshes) before the flowers opened. Nectar volume was measured with a capillary tube (internal diameter 0.3 mm) 1 day after the flower opened [54]. To determine pollen production per flower, five male flowers and five hermaphrodite flowers were collected before anther dehiscence. After being softened in 8 mol/L sodium hydroxide solution for 10 min, all anthers for each flower were crushed and mixed in 1 mL of 10% potassium chloride solution. Next, 10 µL of the mixture was deposited on a microscopic slide and the number of pollen grains was counted under a light microscope. The examination was repeated four times per sample, and the obtained values were averaged. To determine the dry weight of the flowers, the female, male, or hermaphrodite flowers collected 100 flowers separately and then were freeze dried and weighted.

### 4.3. Collection and Analysis of Floral Scent

Flowers from six males, six females, and six hermaphrodites were collected and immediately cooled in liquid nitrogen. In the laboratory, the frozen flowers were ground to a fine powder in liquid nitrogen and stored at −80°C until analysis. The quantity was 1.250 g for each sample and 200 ng 2-methyl-3-heptanone was added as an internal standard. HS-SPME sampling was performed by a CTC auto-sampler. A 50/30 μm DVB/CAR/PDMS (divinylbenzene/carboxen/polydimethylsiloxane) fiber (Sigma-Aldrich, Inc., St. Louis, MI, USA) was used. Samples were incubated for 45 min at 50°C and the SPME fiber was exposed to the headspace for 30 min before desorption for 5 min at 260 °C in the injection port. GC–MS analysis was carried out on an Agilent 7890A/5975C GC-MS platform (Agilent Technologies, Santa Clara, CA, USA). The chromatography was performed using a DB-wax column (30 m × 0.25 mm × 0.25 μm) under the following conditions: splitless; helium as carrier gas at 1 mL/min; injector temperature, 260 °C; oven temperature, started at 40 °C (hold for 5 min), increase of 5 °C/min to 250 °C, and hold for 5 min; electron ionization mode (EI 70 eV); source temperature, 230 °C; quadrupole temperature, 150 °C; scan range, 20~500 a.m.u. The compound tentative identifications were based on the NIST 2011 databases and on comparisons with the existing literature [25,26,27,28,29,30,31,32,33,34,35,36,37,38,39,40,41,42,43,44,45,46,47,48,49].

### 4.4. Pollinator Investigation and Behavioral Test

A pollinator investigation was conducted on sunny days from 08:30~17:00 during November 2015 in wild fields. The specimens of visitors were netted and identified. Four plants of each sex, with similar size, were chosen to record the visiting frequency of the visitors on each plant. Males usually have a larger floral display than females and hermaphrodites. To further investigate the selective preference of pollinators, a controlled experiment was carried out from 12 to 16 November 2016, in which cut branches were used. On sunny days, branches from each sex were collected in the early morning and each was inserted into a glass jar containing purified water. Redundant flowers were removed to ensure each branch had the same number of flowers (about 200). The cut branches were arranged in three rows and three columns 30 cm apart. To avoid the influence of context, a site at least 100 m away from any *E. loquaiana* plants was chosen. Because the branches are less attractive to insects, this experiment was conducted near an artificial hive. To avoid the influence of direction, the branch arrangement was changed every hour. The visiting time of each visitor and the type they chose were recorded.

To determine the response of pollinators to the scent of the different sexes, a four-arm olfactometer (PSM4-300, PUSEN cop., Nanjing, Jiangsu Province, China) was used. Two hundred fresh flowers from each sex were placed in a scent source bottle and then introduced to three of the four arms, respectively. An empty scent source bottle was introduced to the rest arm, which served as a control. Twenty bees (artificial feeding) were introduced to the olfactometer in one test and their choices were recorded after 5 min. The arms were alternated in every test to avoid position effects, and ten tests were conducted in total.

### 4.5. Data Analysis

We used nonlinear multidimensional scaling (NMDS) based on Bray–Curtis similarities to detect similarities in the relative amounts of scent components among sexes. Stress values were reported, with smaller stress values indicating a better fit of the reproduced ordination to the underlying distance matrix [55]. An analysis of similarities (ANOSIM) was then used to test for significant differences between sexes. The above analyses were conducted using R (ver. 3.4.3, R Development Core Team, https://www.r-project.org, accessed on 1 August 2022).

Floral scent data were compared using a one-way analysis of variance (ANOVA), followed by Duncan’s multiple range test. Morphology data were compared using the Kruskal–Wallis ANOVA followed by the Dunn–Bonferroni test or compared by an unpaired *t*-test. The analyses were carried out using SPSS 20.0 (SPSS Inc., Chicago, IL, USA).

## Figures and Tables

**Figure 1 plants-11-02560-f001:**
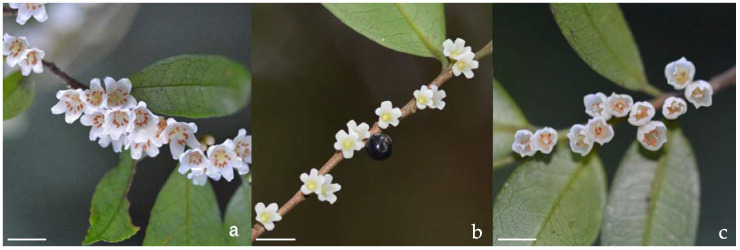
Flowering branches of *Eurya loquaiana* from male (**a**), female (**b**), and hermaphrodite (**c**) plants. The scale bars are 5 mm.

**Figure 2 plants-11-02560-f002:**
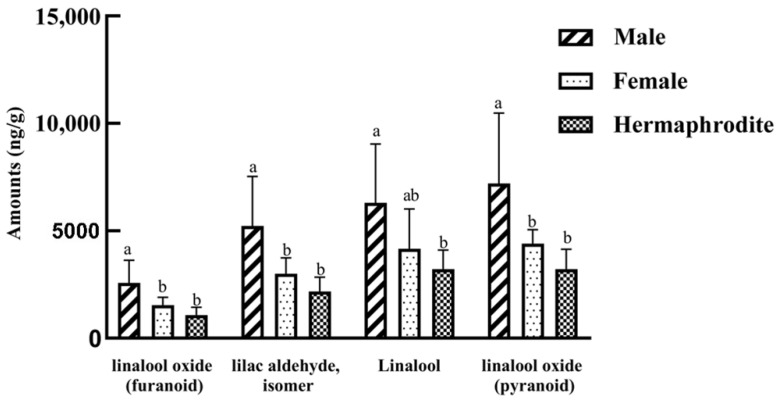
Comparison of major substance amounts in different sexual types. Different lower-case letters (a–c) indicate significant differences among sexes (*p* < 0.05).

**Figure 3 plants-11-02560-f003:**
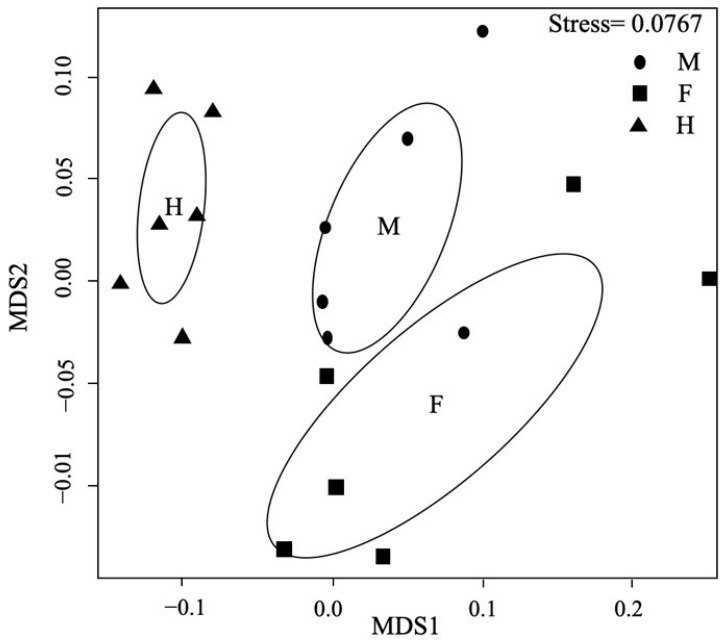
Nonmetric multidimensional scaling (NMDS) of *Eurya loquaiana* floral scent composition. Male (M), female (F), and hermaphrodite (H) flowers were tested.

**Figure 4 plants-11-02560-f004:**
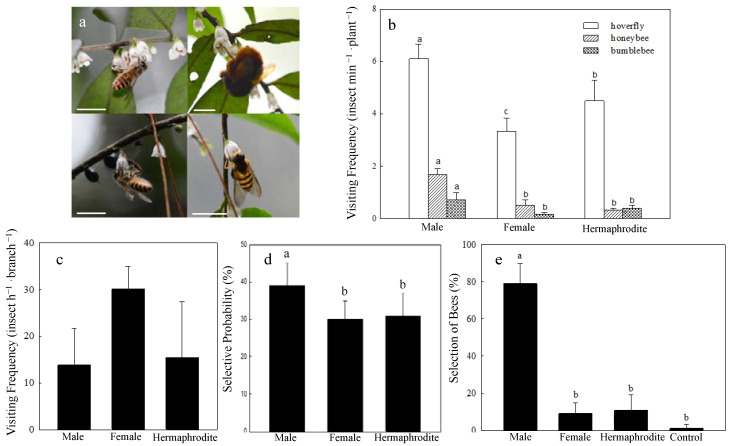
Main visitors of *Eurya loquaiana* (**a**) and their responses to different sexual types (**b**–**e**). (**a**) Honeybee (left), bumblebee (top right), and hoverfly (lower right), the scale bars are 10 mm. (**b**) Visiting frequency of the main visitors in the wild. (**c**) Visiting frequency in the controlled experiment. (**d**) The selective probability of visitors in the controlled experiment. (**e**) Honeybee responses to the floral scent. Different lower-case letters (a–c) above the bars indicate significant differences among sexes or treatments (*p* < 0.05).

**Table 1 plants-11-02560-t001:** Floral morphology and rewards. Comparing male, female, and hermaphrodite flowers.

Characteristic	Male	Female	Hermaphrodite
Petal length (mm)	3.55 ± 0.32 a ^1^	2.80 ± 0.60 b	3.11 ± 0.34 b
Petal width (mm)	1.89 ± 0.18 a	1.39 ± 0.30 c	1.65 ± 0.26 b
Dry weight of 100 flowers (g)	0.3854 ± 0.0372 a	0.2551 ± 0.0411 b	0.2366 ± 0.0922 b
Stamen length (mm)	2.69 ± 0.37 a	-	2.48 ± 0.32 b
Stamen number per flower	10.36 ± 1.56 a	-	5.02 ± 2.63 b
Pollen number per flower	13,750 ± 1041 a	-	4667 ± 702 b
Nectar volume per flower (μL)	-	1.76 ± 0.29 a	0.47 ± 0.24 b

^1^ Values are expressed as mean ± SD, with a different letter (a–c) in the same row indicating a significant difference (*p* < 0.05).

**Table 2 plants-11-02560-t002:** Volatile compounds extracted from male, female, and hermaphrodite flowers of *Eurya loquaiana*.

No.	RT	RI	RI*	Compound Name	Relative Content of VOCs (%)		
					Male	Female	Hermaphrodite
1	3.13078	889.08	888 [25]	Methyl Alcohol	2.50 ± 2.47 a	1.55 ± 1.1 a	1.46 ± 0.95 a
2	4.45895	947.37	940 [26]	2-Pentanone	0.72 ± 0.16 b	0.51 ± 0.26 b	2.72 ± 0.85 a
3	4.47228	947.95	969 [27]	3-Pentanone	0.64 ± 0.14 b	0.56 ± 0.09 b	1.02 ± 0.16 a
4	9.802715	1138.4	1146 [28]	Myrcene	0.48 ± 0.14 a	0.62 ± 0.24 a	0.63 ± 0.15 a
5	10.7644	1167.6	1176 [29]	Limonene	0.32 ± 0.12 ab	0.23 ± 0.07 b	0.4 ± 0.07 a
6	11.61285	1193.5	1216 [30]	Benzene, 1-ethyl-4-methyl-	0.43 ± 0.38 ab	0.49 ± 0.2 a	0.11 ± 0.13 b
7	11.7528	1197.6	1207 [31]	2-Hexenal, isomer	1.99 ± 0.56 a	1.94 ± 0.73 a	0.42 ± 0.19 b
8	12.03705	1206.4	1220 [32]	Furan, 2-pentyl-	0.39 ± 0.19 a	0.18 ± 0.07 b	0.16 ± 0.08 b
9	12.18805	1211.2	1241 [33]	trans-β-Ocimene	0.2 ± 0.1 a	0.25 ± 0.16 a	0.24 ± 0.06 a
10	12.717775	1227.7	1230 [28]	cis-β-Ocimene	0.22 ± 0.13 b	0.42 ± 0.22 a	0.21 ± 0.12 b
11	12.82995	1231.1	1231 [34]	Benzene, 1-ethyl-2-methyl-	0.44 ± 0.11 a	0.29 ± 0.04 b	0.34 ± 0.11 ab
12	13.192	1242.6	1266 [35]	Thiocyanic acid, methyl ester	0.41 ± 0.37 a	0.25 ± 0.14 a	0.3 ± 0.2 a
13	14.227	1275	1273 [36]	Octanal	0.15 ± 0.03 a	0.17 ± 0.09 a	0.01 ± 0.03 b
14	15.1598	1304.5	1304 [37]	2-Penten-1-ol, isomer	0.19 ± 0.12 a	0.15 ± 0.03 a	0.21 ± 0.11 a
15	15.4174	1313.1	1313 [38]	5-Hepten-2-one, 6-methyl-	0.21 ± 0.1 a	0.13 ± 0.09 a	0.24 ± 0.09 a
16	16.1837	1338.7	1339 [39]	1-Hexanol	2.88 ± 0.79 b	2.32 ± 0.48 b	3.75 ± 0.51 a
17	16.96105	1364.8	1365 [40]	3-Hexen-1-ol, isomer	3.7 ± 0.8 b	2.9 ± 0.84 b	7.44 ± 1.12 a
18	17.2342	1374	1373 [41]	Nonanal	0.62 ± 0.39 a	0.45 ± 0.08 a	0.73 ± 0.15 a
19	18.5824	1421.7	1423 [42]	Linalool oxide (furanoid)	9.67 ± 0.29 a	10.07 ± 1.19 a	8.47 ± 0.52 b
20	19.38415	1452.3	-	Lilac aldehyde, isomer	19.54 ± 1.1 a	19.6 ± 2.12 a	17.14 ± 0.8 b
21	21.33425	1528.2	1528 [43]	Linalool	23.32 ± 2.79 a	25.67 ± 6.54 a	25.35 ± 1.95 a
22	24.7146	1670.8	1671 [28]	α-Terpineol	0.64 ± 0.24 a	0.35 ± 0.2 b	0.59 ± 0.22 ab
23	26.1894	1737	1742 [44]	Methyl salicylate	1 ± 0.61 a	0.59 ± 0.27 a	0.71 ± 0.26 a
24	26.336	1743.8	1745 [45]	Linalool oxide (pyranoid)	26.77 ± 1.85 ab	29.06 ± 3.49 a	25.27 ± 0.96 b
25	27.8107	1812.2	1812 [46]	Hexanoic acid	1.14 ± 0.23 a	0.41 ± 0.18 c	0.89 ± 0.17 b
26	28.34375	1837.9	1838 [47]	Geraniol	0.5 ± 0.12 a	0.43 ± 0.14 a	0.46 ± 0.22 a
27	30.2716	1931.7	1938 [48]	2-Hexenoic acid, isomer	0.63 ± 0.35 a	0.2 ± 0.16 b	0.47 ± 0.15 ab
28	34.0873	2125.3	2122 [49]	Eugenol	0.31 ± 0.13 a	0.2 ± 0.05 a	0.27 ± 0.08 a
**Terpenoids**					81.86 ± 3.69 b	86.85 ± 2.47 a	78.97 ± 2.69 b
**Benzenoids**					2.17 ± 0.50 a	1.58 ± 0.36 a	1.43 ± 0.46 a
**Aliphatic hydrocarbons**					15.97 ± 3.51 b	11.57 ± 2.38 c	19.60 ± 2.33 a

Values are expressed as mean ± SD of sextuplicate measurements, with a different letter (a–c) in the same row indicating significant difference according to ANOVA test (*p* < 0.05). RT—retention time; RI—retention index; RI*—retention index from the literature.

## Data Availability

Not applicable.

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
