# Peer review of "Sexual Differences in Eurya loquaiana Dunn Floral Scent and How Pollinators Respond"

_plants, 2022, doi:10.3390/plants11192560_

Round 1

Reviewer 1 Report

Wang et al report on differences in pollinator attraction to female, male and hermaphrodite flowers of Eruya species, and try to explain the differentiation by morphological traits and VOC profiles.

The study seems sound when it comes to the biology, while unfortunately the chemical studies would need to be improved for me to recommend publication.

The methodology of freeze-drying the flowers and later extract with SPME does not make sense. The point with HS analysis is to sample what is being emitted by an organism (alive!). If the material is freeze-dried it would have been much more appropriate to extract in a semi-polar general solvent (such as dichloromethane, diethyl ether etc), which would also be less selective than SPME between different types of compounds. However, even if the logic is lacking, profiles for all samples, and relative qualitative differences were found.

What really needs to be corrected before publication, in my opinion, is the identification of compounds. First, all identifications here are tentative, at best (as no standards were used). At minimum, please provide literature values for all RI in table 2 so that your data can be validated. Those literature values should be from studies where synthetic, structurally confirmed standards were used! Second, please remove all specific identifications, for example "lilac aldehyde C". There is no way you can know that you don't have any of the other isomers with the methodology you present! Similarly, Z/E-isomers of linalool oxides are virtually impossible to tell apart without standards. But, you must report if they are furanoids or pyranoids! In table 2, please remove CAS-numbers as they differ between isomers/enantiomers! Throughout, please specify that all identifications are tentative and remove all specific data from table 2 that you don't have proof for, and instead report "linalool oxide, isomer" etc.

Some minor corrections>

P2L61 "forging" should be "foraging"

P3L80 italise taxon, add "tentative" before "identification"

P3L86 add "were"

P4L113-114, awkward sentence, please rephrase

P4L115 change "sucked" to "fed from", or similar

P4L123 change "produced" to "extracted from"

Table 2 - edit as above, add RI ref, tentative ID, remove CAS, remove specifics that are not confirmed

P5L142 change "dominated by" to "common" or otherwise rephrase.

P6L161 change "flora" to "floral"

P6L162 "this" needs to be specified

P6L168-170 please rephrase, awkward sentence

P6L171-173 please rephrase, these bees are not greater than all others, point out that they are good learners without saying they are the best.

Reviewer 2 Report

Dear authors,

This is a manuscript that refers to research on a little-addressed topic, such as the composition of essence according to sex in plants.

Thus, the manuscript reflects an investigation with that intention and with some aspects that, in my opinion, should be revised.

For example, why in the second paragraph of the Introduction, the citations are not highlighted in blue? Appointments (2,3), (8-12) and (13-15);

In section 2.3, in comparison of the floral scent, line 87, reference is made to table 1, however, the information corresponds to Table 2.

In Discussion, there is confusion between what is described in lines 137 and 138, with respect to the research that is cited, since it refers to the detection of 87 compounds from flower buds and this work was carried out on flowers and aerial parts, such as leaves and stems (Motooka et al, 2015), which later, lines 150 and 151 is confirmed in the description.

In Line 154, it is indicated that in this study a similar composition was found in terms of the presence of alpha - pinene in male, female and hermaphrodite flowers (according to table 2) as reported by Miyazawa et al (20) and Wang et al (19). However, in the table of this research, this compound (a-pinene) is not reported and stands out as a possible chemical agent involved in the attraction to pollinators, in male flowers. Here it should be noted that in methodology, it is described that a DB-WAX composition was used as a chromatographic column for GC-MS, which is reported in the papers by Motooka and Wang, but with medium polarity fused silica (HP5-MS).

Reviewer 3 Report

General comments:

The present manuscript describes the investigation of floral scent components of male, female and hermaphrodite flowers and their effect on pollinator behavior from the species Eurya loquaiana Dunn.  

Results demonstrate the importance of scent profile for pollinator attractiveness and how small differences in the relative scent profile may affect pollination. However, the authors do not present differences in absolute quantity of scent production, which may be an important factor affecting foraging behavior and should be included in this study.

Detailed comments:

Abstract:

L17 (L61): forging behavior – change to foraging

L19 (L83): main aroma-active compounds-please shortly mention the criteria (by frequency, most consistent among all flower types…)

Introduction:

Please add some examples/references stating on the effect of differences in floral scent between male and female flowers on pollinator foraging behavior.

Results:

L 82: this should be table 2

Paragraph 2.2 – as already stated for the abstract, authors should state more in detail how they consider the most aroma active compounds. I think it is essential to compare absolute quantities of scent compounds produced by the different flower types (as mentioned by the authors in the discussion) in order to make a final statement about the significance of VOC emission among the different flower types.

L113: The Hoverfly’s smooth body reduces the pollination efficiency of the pollen – please give reference or own data for this statement.

Paragraph 2.4 – for better understanding, authors should include the most outstanding data from Fig. 3 within the text (i.e. percentage difference in flower visitation among flower types). The procedure of the selective experiment “c” (Fig. 3) should be mentioned here shortly.

Discussion:

L160 – this would be Fig. 3 c-d

Authors should discuss more in detail the strong differences in insect visits between male and hermaphrodite flowers despite the high similarity in VOC aroma profile and relative high pollen award.

Material and Methods:

L195: how many flowers were analyzed for pollen amount?

L202: Three samples were determined for each sex? -what does this refer to?

L221:from (not form)

Round 2

Reviewer 1 Report

The authors have corrected most issues with the previous version. There are a few things that still need to be fixed before publication, I think.

The language in the new additions needs some help - please have this proof read by a native English speaker.

Abstract

Add "tentatively" before "identified" (twice)

L18 "scents" to "scent"

L20 remove "isomer" (sufficient that is in table 2)

L21 Remove "...whose relative content...flowers." Not needed in abstract.

P3 

L90 "aliphatics" to "aliphatic hydrocarbons" (if that is what you mean). 

"terpenes" to "terpenoids" as you later refer to a sub group as terpenoids, i.e. they are not all terpenes?

L91 What do you mean by "the main aroma-active". You have not tested for that?! Do you mean "most abundant"?!

L93 Remove "isomer", "were" should be "was".

L97-102: Reduce to one decimal point in these percentages considering the large variation.

L104 remove "isomer"

L106-107 Your IS is "2-methyl-3-heptanone". Do not write like it is a table!

L107 Remove "were"

L114 Change "obvious" (subjective) to "clear" (objective)

P4

L123 "were" to "was"

L125 "is" to "was"

L128 "sucking" to "feeding"

L129 "pollens" to "pollen"

L130 "show" to "showed", "sucking" to "feeding of"

L135 "exist" to "exists"

Table 2: Heading - rephrase: You sampled with SPME and analysed with GC-MS. Not coupled!

IMPORTANT now you have added RI values from the literature - then please use them!!! All values that are more than 20-30 off mean you have misidentified that compound (or lit. value is incorrect). Please double check all those, and remove ID for compounds that do not match what you believe are reliable lit. values (and please make sure all lit. values are from SYNTHETIC STANDARDS, and not other authors best guesses!)

Number 13 should be cis-ocimene? (As you have specified trans already)

Remove comments about CAS numbers.

Discussion

L176 remove "isomer"

L180 "under" to "in"

L182 "scents are" to "scent is"

L184-190. Rephrase. Differences are not due to different detection methods (what were they by the way?) but due to different sample types!

L198 "has" to "have"

L204 "payment" to "reward"?

L216. You say that you further confirm that your system is an intermediate evolutionary stage. How can you do that? Who confirmed it in the first place? And your result can not possible confirm such a thing - but merely "suggest" or "indicate" that this is true. Please do not overstate the implications of your results! 

Methods

L252 remove "injection"

L253 write the standard in non-table format

L264 add "tentative"

Reviewer 3 Report

The authors related satisfactorily to the reviewer comments.

Author Response

Thank you for your affirmation.